# Linear Relaxations for Finding Diverse Elements in Metric Spaces

**Aditya Bhaskara**
University of Utah
bhaskara@cs.utah.edu

**Mehrdad Ghadiri**
Sharif University of Technology
ghadiri@ce.sharif.edu

**Vahab Mirrokni**
Google Research
mirrokni@google.com

**Ola Svensson**
EPFL
ola.svensson@epfl.ch

## Abstract

Choosing a diverse subset of a large collection of points in a metric space is a fundamental problem, with applications in feature selection, recommender systems, web search, data summarization, etc. Various notions of diversity have been proposed, tailored to different applications. The general algorithmic goal is to find a subset of points that maximize diversity, while obeying a cardinality (or more generally, matroid) constraint. The goal of this paper is to develop a novel linear programming (LP) framework that allows us to design approximation algorithms for such problems. We study an objective known as *sum-min* diversity, which is known to be effective in many applications, and give the first constant factor approximation algorithm. Our LP framework allows us to easily incorporate additional constraints, as well as secondary objectives. We also prove a hardness result for two natural diversity objectives, under the so-called *planted clique* assumption. Finally, we study the empirical performance of our algorithm on several standard datasets. We first study the approximation quality of the algorithm by comparing with the LP objective. Then, we compare the quality of the solutions produced by our method with other popular diversity maximization algorithms.

## 1 Introduction

Computing a concise, yet *diverse* and *representative* subset of a large collection of elements is a central problem in many areas. In machine learning, it has been used for feature selection [23], and in recommender systems [24]. There are also several data mining applications, such as web search [21, 20], news aggregation [2], etc. Diversity maximization has also found applications in drug discovery, where the goal is to choose a small and diverse subset of a large collection of compounds to use for testing [16].

A general way to formalize the problem is as follows: we are given a set of objects in a metric space, and the goal is to find a subset of them of a prescribed size so as to maximize some measure of diversity (a function of the distances between the chosen points). One well studied example of a diversity measure is the minimum pairwise distance between the selected points – the larger it is, the more "mutually separated" the chosen points are. This, as well as other diversity measures have been studied in the literature [11, 10, 6, 23], including those based on mutual information and linear algebraic notions of distance, and approximation algorithms have been proposed. This is similar in spirit to the rich and beautiful literature on clustering problems with various objectives (e.g. $k$-center, $k$-median, $k$-means). Similar to clustering, many of the variants of diversity maximization admit

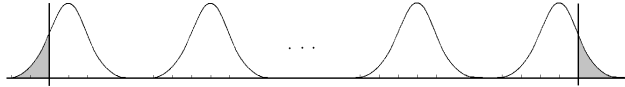

Figure 1: Preference for far clusters in sum-sum($\cdot$) maximization

constant factor approximation algorithms. Most of the known algorithms for diversity maximization are based on a natural greedy approach, or on local search.

Our goal in this work is to develop novel linear programming formulations for diversity maximization and provide new approximation guarantees. Convex relaxation approaches are typically powerful in that they can incorporate additional constraints and additional objective functions, as we will illustrate. This is important in some applications, and indeed, diversity maximization has been studied under additional knapsack [3] and matroid [2] constraints. In applications such as web search, it is important to optimize diversity, *along with* other objectives, such as total relevance or coverage (see [4]). Another contribution of this work is to explore approximation lower bounds for diversity maximization. Given the simplicity of the best known algorithms for some objectives (e.g., greedy addition, single-swap local search), it is natural to ask if better algorithms are possible. Rather surprisingly, we show that the answer is no, for the most common objectives.

**Objective functions.** The many variants of diversity maximization differ in their choice of the objective function, i.e., how they define diversity of a set $S$ of points. Our focus in this paper will be *distance based* objectives, which can be defined over arbitrary metric spaces, via pairwise distances between the chosen points. Let $d(u, v)$ be the distance between points $u$ and $v$, and for a set of points $T$, let $d(u, T) = \min_{v \in T} d(u, v)$. The three most common objectives are:

1. *Min-min diversity*, defined by min-min($S$) = $\min_{u \in S} d(u, S \setminus u)$.
2. *Sum-min diversity*, defined by sum-min($S$) = $\sum_{u \in S} d(u, S \setminus u)$.
3. *Sum-sum diversity*, defined by sum-sum($S$) = $\sum_{u \in S} \sum_{v \in S} d(u, v)$.

All three objectives have been used in applications [16]. Of these min-min and sum-sum are also known to admit constant factor approximation algorithms. In fact, a natural greedy algorithm gives a factor $1/2$ approximation for min-min, while local search gives a constant factor approximation for sum-sum, even with matroid constraints [6, 2, 4]. However, for the *sum-min* objective, the best known algorithm had an approximation factor of $O(1/\log n)$ [6] and no inapproximability results were known. Combinatorial methods such as greedy and local search fail (see Lemma 1), and achieving a constant factor approximation has remained a challenge. Compared to the other objectives, the sets that maximize the sum-min objective have properties that are desirable in practice, as observed in [16], and demonstrated in our experiments. We will now outline some theoretical reasons.

**Drawbacks of the *min-min* and *sum-sum* objectives.** The main problem with min-min stems from the fact that it solely depends on the closest pair of chosen points, and it does not capture the distance distribution between the chosen points well. Another concern is that it is highly non-monotone in the size of $|S|$ – in applications such as search, it is paradoxical for the diversity to take a sharp drop once we add one extra element to the set of search results. The sum-sum objective is much more robust, and is hence much more popular in applications. However, as also noted in [16], it tends to promote picking too many "corner" points. To illustrate, suppose we have a set of points that fall into $k$ clusters (which is common in candidate search results). Suppose the points are distributed as a mixture of $k$ equally spaced Gaussians on a line (see Figure 1). The intuitively desired solution is to pick one point from each of the clusters. However the optimizer for sum-sum picks all the points from the farthest two clusters (shown shaded in Figure 1).

The sum-min objective inherits the good properties of both – it is robust to a small number of additions/removals, *and* it tries to ensure that each point is far from the others. However, it is trickier to optimize, as we mentioned earlier. In fact, in the supplement, Section E, we show that:

**Lemma 1.** *The natural Greedy and Local-Search algorithms for sum-min diversity have an approximation ratio of $O(1/\sqrt{k})$.*

**Our contributions.** With these motivations, we study the problem of maximizing sum-min diversity subject to a cardinality constraint – $\max_{|S| \leq k}$ sum-min$(S)$. Our main algorithmic results are:

- We give a factor $1/8$ approximation for sum-min diversity maximization with cardinality constraint (the first constant factor approximation). Indeed, when $k$ is a large enough constant, we give a (roughly) $\frac{1}{2e}$-approximation. This is presented in Section 2 to illustrate our ideas (Theorem 1). The algorithm can also incorporate arbitrary concave functions of distance, as well as explicit constraints to avoid duplicates (end of Section 2).

- We show that the $1/8$ approximation holds when we replace cardinality constraints with arbitrary matroid constraints. Such constraints arise in applications such as product search [3] or news aggregators [2] where it is desirable to report items from different brands or different news agencies. This can be modeled as a partition matroid.

- Our formulation can be used to maximize the sum-min diversity, along with *total relevance* or *coverage* objectives (Theorem 3). This is motivated by applications in recommender systems in which we also want the set of results we output to *cover* a large range of topics [4, 2], or have a high total relevance to a query.

Next, we show that for both the sum-sum and the sum-min variants of diversity maximization, obtaining an approximation factor better than $1/2$ is hard, under the planted clique assumption (Theorem 5). (We observe that such a result for min-min is easy, by a reduction from independent set.) This implies that the simple local search algorithms developed for the sum-sum diversity maximization problem [6, 10, 11] are the best possible under the planted clique assumption.

Finally, we study the empirical performance of our algorithm on several standard datasets. Our goal here is two-fold: first, we make an experimental case for the sum-min objective, by comparing the quality of the solutions output by our algorithm (which aims to maximize sum-min) with other popular algorithms (that maximize sum-sum). This is measured by how well the solution covers various clusters in the data, as well as by measuring quality in a feature selection task. Second, we study the approximation quality of the algorithm on real datasets, and observe that it performs much better than the theoretical guarantee (factor $1/8$).

## 1.1 Notation and Preliminaries

Throughout, $(V, d)$ will denote the metric space we are working with, and we will write $n = |V|$. The number of points we need to output will, unless specified otherwise, be denoted by $k$.

**Approximation factor.** We say that an algorithm provides an $\alpha$ factor approximation if, on every instance, it outputs a solution whose objective value is at least $\alpha \cdot$ opt, where opt is the optimum value of the objective. (Since we wish to maximize our diversity objectives, $\alpha$ will be $\leq 1$, and ratios closer to 1 are better.)

**Monotonicity of sum-min.** We observe that our main objective, sum-min$(\cdot)$, is not monotone. I.e., sum-min$(S \cup u)$ could be $\leq$ sum-min$(S)$ (for instance, if $u$ is very close to one of the elements of $S$). This means that it could be better for an algorithm to output $k' < k$ elements if the goal is to maximize sum-min$(\cdot)$. However, this non-monotonicity is not too serious a problem, as the following lemma shows (proof in the supplement, Section A.1).

**Lemma 2.** *Let $(V, d)$ be a metric space, and $n = |V|$. Suppose $1 < k < n/3$ be the target number of elements. Let $S'$ be any subset of $V$ of size $\leq k$. Then we can efficiently find an $S \subseteq V$ of size $= k$, such that* sum-min$(S) \geq 1/4 \cdot$ sum-min$(S')$.

Since our aim is to design a constant factor approximation algorithm, in what follows, we will allow our algorithms to output $\leq k$ elements (we can then use the lemma above to output precisely $k$).

**Matroid constraints.** Let $D$ be a ground set of elements (which in our case, it will be $V$ or its subset). A matroid $\mathcal{M}$ is defined by $\mathcal{I}$, a family of subsets of $D$, called the *independent sets* of the matroid. $\mathcal{I}$ is required to have the properties of being *subset-closed* and having the *basis exchange* property (see Schrijver [22] for details). Some well-studied matroids which we consider are: (a) the uniform matroid of rank $k$, for which we have $\mathcal{I} := \{X \subseteq D : |X| \leq k\}$, (b) partition matroids, which are the direct sum of uniform matroids.

In matroid constrained diversity maximization, we are given a matroid $\mathcal{M}$ as above, and the goal is to output an element of $\mathcal{I}$ that maximizes diversity. Note that if $\mathcal{M}$ is the uniform matroid, this is equivalent to a cardinality constraint. The *matroid polytope* $P(\mathcal{M})$, defined to be the convex hull of the indicator vectors of sets in $\mathcal{I}$, plays a key role in optimization under matroid constraints. For most matroids of practical interest, it turns out optimization over $P(\mathcal{M})$ can be done in polynomial time.

## 2 Basic Linear Programming Formulation

We will now illustrate the main ideas behind our LP framework. We do so by proving a slightly simpler form of our result, where we assume that $k$ is *not too small*. Specifically, we show that:

**Theorem 1.** *Let $(V, d)$ be a metric space on $n$ points, and let $\epsilon, k$ be parameters that satisfy $\epsilon \in (0, 1)$ and $k > 8\log(1/\epsilon)/\epsilon^2$. There is a randomized polynomial time algorithm that outputs a set $S \subseteq V$ of size $\leq k$ with $\mathbb{E}[\textsf{sum-min}(S)] \geq \frac{1-2\epsilon}{2e} \cdot \textsf{opt}$, where $\textsf{opt}$ is the largest possible $\textsf{sum-min}()$ value for a subset of $V$ of size $\leq k$.*

The main challenge in formulating an LP for the $\textsf{sum-min}$ objective is to capture the quantity $d(u, S \setminus u)$. The key trick is to introduce new variables to do so. To make things formal, for $i \in V$, we denote by $R_i = \{d(i,j) \ : \ j \neq i\}$ the set of candidate distances from $i$ to its closest point in $S$. Next, let $B(i, r)$ denote the "open" ball of radius $r$ centered at $i$, i.e., $B(i, r) = \{j \in V \ : \ d(i, j) < r\}$; and let $B'(i, r) = B(i, r/2)$ denote the ball of half the radius.

The LP we consider is as follows: we have a variable $x_{ir}$ for each $i \in V$ and $r \in R_i$ which is supposed to be 1 iff $i \in S$ and $r = \min_{j \in S \setminus \{i\}} d(i, j)$. Thus for every $i$, at most one $x_{ir}$ is 1 and the rest are 0. Hence $\sum_{i, r \in R_i} x_{ir} \leq k$ for the intended solution. The other set of constraints we add is the following: for each $u \in V$,

$$\sum_{i \in V, r \in R_i : u \in B'(i,r)} x_{ir} \leq 1. \qquad \text{(figure in Section A.3 of supplement)} \qquad (1)$$

These constraints are the crux of our LP formulation. They capture the fact that if we take any solution $S \subseteq V$, the balls $B(s, r/2)$, where $s \in S$ and $r = d(s, S \setminus \{s\})$ are disjoint. This is because if $u \in B'(i_1, r_1) \cap B'(i_2, r_2)$, then assuming $r_1 \geq r_2$ (w.l.o.g.), triangle inequality implies that $d(i_1, i_2) < r_1$ (the strict inequality is because we defined the balls to be 'open'); Thus, in an integral solution, we will set at most one of $x_{i_1 r_1}$ and $x_{i_2 r_2}$ to 1. The full LP can now be written as follows

$$\text{maximize} \ \sum_i \sum_{r \in R_i} x_{ir} \cdot r \quad \text{subject to}$$

$$\sum_{i \in V, r \in R_i} x_{ir} \leq k,$$

$$\sum_{i \in V, r \in R_i : u \in B'(i,r)} x_{ir} \leq 1 \qquad \text{for all } u \in V,$$

$$0 \leq x_{ir} \leq 1.$$

The algorithm then proceeds by solving this LP, and rounding via the procedure defined below. Note that after step 2, we may have pairs with the same first coordinate, since we round them independently. But after step 3, this will not happen, as all but one of them will have been removed.

---

procedure $\textsf{round}(x)$     // LP solution $(x)$
1: Initialize $\mathcal{S} = \emptyset$.
2: Add $(i, r)$ to $\mathcal{S}$ with probability $(1 - \epsilon)(1 - e^{-x_{ir}})$ (independent of the other point-radius pairs).
3: If $(i, r) \neq (j, r') \in \mathcal{S}$ such that $r \leq r'$ and $i \in B'(j, r')$, remove $(i, r)$ from $\mathcal{S}$.
4: If $|\mathcal{S}| > k$, abort (i.e., return $\emptyset$ which has value 0); else return $S$, the set of first coordinates of $\mathcal{S}$.

---

**Running time.** The LP as described contains $n^2$ variables, $n$ for each vertex. This can easily be reduced to $O(\log n)$ per vertex, by only considering $r$ in multiples of $(1 + \delta)$, for some fixed $\delta > 0$.

Further, we note that the LP is a *packing LP*. Thus it can be solved in time that is nearly linear in the size (and can be solved in parallel) [19].

**Analysis.** Let us now show that round returns a solution to with large expected value for the objective (note that due to the last step, it always returns a *feasible* solution, i.e., size $\leq k$). The idea is to write the expected diversity as a sum of appropriately defined random variables, and then use the linearity of expectation. For a (vertex, radius) pair $(i, r)$, define $\chi_{ir}$ to be an indicator random variable that is 1 iff (a) the pair $(i, r)$ is picked in step 2, (b) it is not removed in step 3, and (c) $|\mathcal{S}| \leq k$ after step 3. Then we have the following.

**Lemma 3.** *Let $S$ be the solution output by the algorithm, and define $\chi_{ir}$ as above. Then we have* sum-min$(S) \geq \sum_{i,r} \frac{r}{2} \cdot \chi_{ir}$.

*Proof.* If the set $\mathcal{S}$ after step 3 is of size $> k$, each $\chi_{ir} = 0$, and so there is nothing to prove. Otherwise, consider the set $\mathcal{S}$ at the end of step 3 and consider two pairs $(i, r), (j, r') \in \mathcal{S}$. The fact that both of them survived step 3 implies that $d(i, j) \geq \max(r, r')/2$. Thus $d(i, j) \geq r/2$ for any $j \neq i$ in the output, which implies that the contribution of $i$ to the sum-min objective is $\geq r/2$. This completes the proof. $\square$

Now, we will fix one pair $(i, r)$ and show a lower bound on $\Pr[\chi_{ir} = 1]$.

**Lemma 4.** *Consider the execution of the algorithm, and consider some pair $(i, r)$. Define $\chi_{ir}$ as above. We have $\Pr[\chi_{ir} = 1] \geq (1 - 2\epsilon)x_{ir}/e$.*

*Proof.* Let $T$ be the set of all (point, radius) pairs $(j, r')$ such that $(i, r) \neq (j, r')$, $i \in B'(j, r')$, and $r' \geq r$. Now, the condition (b) in the definition of $\chi_{ir}$ is equivalent to the condition that none of the pairs in $T$ are picked in step 2. Let us denote by $\chi^{(a)}$ (resp., $\chi^{(b)}, \chi^{(c)}$) the indicator variable for the condition (a) (resp. (b), (c)) in the definition of $\chi_{ir}$. We need to lower bound $\Pr[\chi^{(a)} \wedge \chi^{(b)} \wedge \chi^{(c)}]$. To this end, note that

$$\Pr[\chi^{(a)} \wedge \chi^{(b)} \wedge \chi^{(c)}] = \Pr[\chi^{(a)} \wedge \chi^{(b)}] - \Pr[\chi^{(a)} \wedge \chi^{(b)} \wedge \overline{\chi^{(c)}}]$$
$$\geq \Pr[\chi^{(a)} \wedge \chi^{(b)}] - \Pr[\chi^{(a)} \wedge \overline{\chi^{(c)}}]. \tag{2}$$

Here $\overline{\chi^{(c)}}$ denotes the complement of $\chi^{(c)}$, i.e., the event $|\mathcal{S}| > k$ at the end of step 3. Now, since the rounding selects pairs independently, we can lower bound the first term as

$$\Pr[\chi^{(a)} \wedge \chi^{(b)}] \geq (1 - \epsilon)\left(1 - e^{-x_{ir}}\right) \prod_{(j,r') \in T} \left[1 - (1 - \epsilon)(1 - e^{-x_{jr'}})\right]$$
$$\geq (1 - \epsilon)\left(1 - e^{-x_{ir}}\right) \prod_{(j,r') \in T} e^{-x_{jr'}} \tag{3}$$

Now, we can upper bound $\sum_{(j,r') \in T} x_{jr'}$, by noting that for all such pairs, $B'(j, r')$ contains $i$, and thus the LP constraint for $i$ implies that $\sum_{(j,r') \in T} x_{jr'} \leq 1 - x_{ir}$. Plugging this into (3), we get

$$\Pr[\chi^{(a)} \wedge \chi^{(b)}] \geq (1 - \epsilon)\left(1 - e^{-x_{ir}}\right) e^{-(1 - x_{ir})} = (1 - \epsilon)\frac{e^{x_{ir}} - 1}{e} \geq (1 - \epsilon)x_{ir}/e.$$

We then need to upper bound the second term of (2). This is done using a Chernoff bound, which then implies the lemma. (see the Supplement, Section A.2 for details). $\square$

*Proof of Theorem 1.* The proof follows from Lemmas 3 and 4, together with linearity of expectation. For details, see Section A.3 of the supplementary material. $\square$

**Direct Extensions.** We mention two useful extensions that follow from our argument.
(1) We can explicitly prevent the LP from picking points that are too close to each other (near duplicates). Suppose we are only looking for solutions in which every pair of points are at least a distance $\tau$. Then, we can modify the set of 'candidate' distances $R_i$ for each vertex to only include those $\geq \tau$. This way, in the final solution, all the chosen points are at least $\tau/2$ apart.
(2) Our approximation guarantee also holds if the objective has any monotone concave function $g()$ of $d(u, S \setminus u)$. In the LP, we could maximize $\sum_i \sum_{r \in R_i} x_{ir} \cdot g(r)$, and the monotone concavity (which implies $g(r/2) \geq g(r)/2$) ensures the same approximation ratio. In some settings, having a cap on a vertex's contribution to the objective is useful (e.g., bounding the effect of outliers).

## 3 General Matroid Constraints

Let us now state our general result. It removes the restriction on $k$, and has arbitrary matroid constraints, as opposed to cardinality constraints in Section 2.

**Theorem 2.** *Let $(V, d)$ be a metric space on $n$ points, and let $\mathcal{M} = (V, \mathcal{I})$ be a matroid on $V$. Then there is an efficient randomized algorithm[1] to find an $S \in \mathcal{I}$ whose expected sum-min$(S)$ value is at least opt$/8$, where opt $= \max_{I \in \mathcal{I}}$ sum-min$(I)$.*

The algorithm proceeds by solving an LP relaxation as before. The key differences in the formulation are: (1) we introduce new *opening* variables $y_i := \sum_{r \in R_i} x_{ir}$ for each $i \in V$, and (2) the constraint $\sum_i y_i \leq k$ (which we had written in terms of the $x$ variables) is now replaced with a *general* matroid constraint, which states that $y \in P(\mathcal{M})$. See Section B (of the supplementary material) for the full LP.

This LP is now rounded using a different procedure, which we call generalized-round. Here, instead of independent rounding, we employ the *randomized swap rounding* algorithm (or the closely related *pipage rounding*) of [7], followed by a randomized rounding step.

---
procedure generalized-round$(y, x)$     // LP solution $(y, x)$.
 1: Initialize $S = \emptyset$.
 2: Apply randomized swap rounding to the vector $y/2$ to obtain $Y \in \{0, 1\}^V \cap P(\mathcal{M})$.
 3: For each $i$ with $Y_i = 1$, add $i$ to $S$ and sample a radius $r_i$ according to the probability distribution that selects $r \in R_i$ with probability $x_{ir}/y_i$.
 4: If $i \in B'(j, r_j)$ with $i \neq j \in S$ and $r_j \geq r_i$, remove $i$ from $S$.
 5: Return $S$.

---

Note that the rounding outputs $S$, along with an $r_i$ value for each $i \in S$. The idea behind the analysis is that this rounding has the same properties as randomized rounding, while ensuring that $S$ is an independent set of $\mathcal{M}$. The details, and the proof of Theorem 2 are deferred to the supplementary material (Section B).

## 4 Additional Objectives and Hardness

The LP framework allows us to incorporate "secondary objectives". As an example, consider the problem of selecting search results, in which every candidate page has a *relevance* to the query, along with the metric between pages. Here, we are interested in selecting a subset with a high total relevance, in addition to a large value of sum-min$()$. A generalization of relevance is *coverage*. Suppose every page $u$ comes with a set $C_u$ of *topics* it covers. Now consider the problem of picking a set $S$ of pages so as to simultaneously maximize sum-min$()$ and the *total coverage*, i.e., the size of the union $\cup_{u \in S} C_u$, subject to cardinality constraints. (Coverage generalizes relevance, because if the sets $C_u$ are all disjoint, then $|C_u|$ acts as the relevance of $u$.)

Because we have a simple formulation and rounding procedure, we can easily incorporate a coverage (and therefore relevance) objective into our LP, and obtain *simultaneous* guarantees. We prove the following: (A discussion of the theorem and its proof are deferred to Section C.)

**Theorem 3.** *Let $(V, d)$ be a metric space and let $\{C_u : u \in V\}$ be a collection of subsets of a universe $[m]$. Suppose there exists a set $S^* \subseteq V$ of size $\leq k$ with sum-min$(S^*) =$ opt, and $|\cup_{u \in S^*} C_u| = C$. Then there is an efficient randomized algorithm that outputs a set $S$ satisfying: (1) $\mathbb{E}[|S|] \leq k$, (2) $\mathbb{E}[$sum-min$(S)] \geq$ opt$/8$, and (3) $\mathbb{E}[|\cup_{u \in S} C_u|] \geq C/16$.*

### 4.1 Hardness Beyond Factor $1/2$

For diversity maximization under both the *sum-sum* and the *sum-min* objectives, we show that obtaining approximation ratios better than 2 is unlikely, by a reduction from the so-called *planted clique* problem. Such a reduction for *sum-sum* was independently obtained by Borodin et al. [4]. For completeness, we provide the details and proof in the supplementary material (Section D).

# 5  Experiments

**Goals and design.**   The goal of our experiments is to evaluate the sum-min objective as well as the approximation quality of our algorithm on real datasets. For the first of the two, we consider the $k$-element subsets obtained by maximizing the sum-min objective (using a slight variant of our algorithm), and compare their *quality* (in terms of being representative of the data) with subsets obtained by maximizing the sum-sum objective, which is the most commonly used diversity objective. Since measuring the *quality* as above is not clearly defined, we come up with two measures, using datasets that have a known clustering:

(1) First, we see how well the different clusters are *represented* in the chosen subset. This is important in web search applications, and we do this in two ways: (a) by measuring the number of distinct clusters present, and (b) by observing the "non-uniformity" in the number of nodes picked from the different clusters, measured as a deviation from the mean.

(2) Second, we consider feature-selection. Here, we consider data in which each object has several features, and then we pick a subset of the *features* (treating each feature as a vector of size equal to the number of data points). Then, we restrict data to just the chosen features, and see how well 3-NN clustering in the obtained space (which is much faster to perform than in the original space, due to the reduced number of features) compares with ground-truth clustering.

Let us go into the details of (1) above. We used two datasets with ground-truth clusterings. The first is COIL100, which contains images of 100 different objects [17]. It includes 72 images per object. We convert them into $32 \times 32$ grayscale images and consider 6 pictures per object. We used Euclidean distance as the metric. The second dataset is CDK2 – a drug discovery dataset publicly available in BindingDB.org [15, 1]. It contains 2253 compounds in 151 different clusters. Tanimoto distance, which is widely used in the drug discovery literature (and is similar to Jaccard distance), was used as the metric. Figure 2 (top) shows the number of distinct clusters picked by algorithms for the two objectives, and (bottom) shows the *non-uniformity* in the #(elements) picked from the different clusters (mean std deviation). We note that throughout this section, ***augmented LP*** is the algorithm that first does our LP rounding, and then adds nodes in a greedy manner to as to produce a subset of size precisely $k$ (since randomized rounding could produce smaller sets).

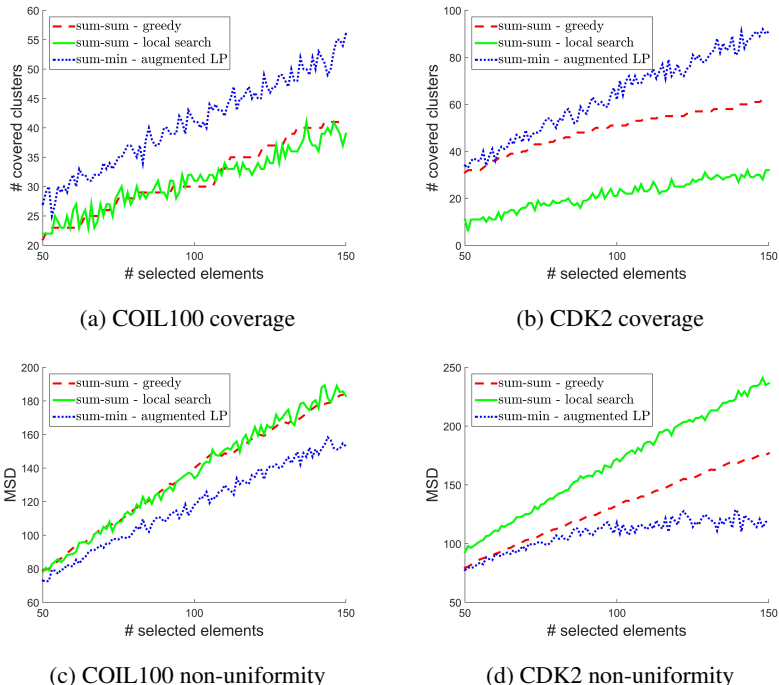

(a) COIL100 coverage  (b) CDK2 coverage

(c) COIL100 non-uniformity  (d) CDK2 non-uniformity

Figure 2: Sum-min vs Sum-sum objectives – how chosen subsets represent clusters

Now consider (2) above – feature selection. We used two handwritten text datasets. Multiple Features is a dataset of handwritten digits (649 features, 2000 instances [14]). USPS is a dataset of handwritten text (256 features, 9298 instances [12, 5]). We used the Euclidean distance as the metric (we could use more sophisticated features to compute distance, but even the simplistic one produces good results). Figure 3 shows the performance of the features selected by various algorithms.

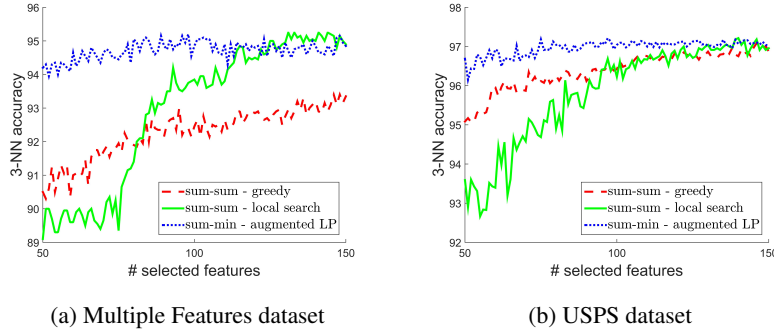

(a) Multiple Features dataset             (b) USPS dataset

Figure 3: Comparing outputs of feature selection via 3-NN classification with 10-fold cross validation.

Next, we evaluate the practical performance of our LP algorithm, in terms of the proximity to the optimum objective value. Since we do not know the optimum, we compare it with the minimum of two upper bounds: the first is simply the value of the LP solution. The second is obtained as follows. For every $i$, let $t_i^j$ denote the $j$th largest distance from $i$ to other points in the dataset. The sum of $k$ largest elements of $\{t_i^{k-1}|i = 1, \dots, n\}$ is clearly an upper bound on the sum-min objective, and sometimes it could be better than the LP optimum. Figure 4 shows the percentage of the minimum of the upperbounds that the augmented-LP algorithm achieves for two datasets [14, 18, 12, 8]. Note that it is significantly better than the theoretical guarantee $1/8$. In fact, by adding the so-called *clique constraints* on the LP, we can obtain an even better bounds on the approximation ratio. However, this will result in a quadratic number of constraints, making the LP approach slow. Figure 4 also depicts the value of the simple LP algorithm (without augmenting to select $k$ points).

Finally, we point out that for many of the datasets we consider, there is no significant difference between the LP based algorithm, and the Local Search (and sometimes even the Greedy) heuristic in terms of the sum-min objective value. However, as we noted, the heuristics do not have worst case guarantees. A comparision is shown in Figure 4 (c).

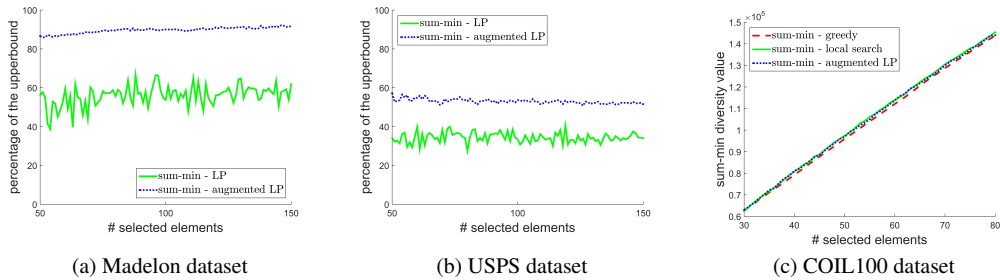

(a) Madelon dataset             (b) USPS dataset             (c) COIL100 dataset

Figure 4: (a) and (b) show the approximation factor of LP and augmented LP algorithms; (c) compares Augmented LP with Greedy and LocalSearch in terms of sum-min objective value

**Conclusions.** We have presented an approximation algorithm for diversity maximization, under the *sum-min* objective, by developing a new linear programming (LP) framework for the problem. Sum-min diversity turns out to be very effective at picking representatives from clustered data – a fact that we have demonstrated experimentally. Simple algorithms such as Greedy and Local Search could perform quite badly for sum-min diversity, which led us to the design of the LP approach. The approximation factor turns out to be much better in practice (compared to $1/8$, which is the theoretical bound). Our LP approach is also quite general, and can easily incorporate additional objectives (such as *relevance*), which often arise in applications.

## Footnotes

[1]Assuming optimization over $P(\mathcal{M})$ can be done efficiently, which is true for all standard matroids.

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
