[Supplementary Material]

# Supplementary Material

## A  Full proofs of Lemmas

### A.1  Proof of Lemma 2

Let us suppose $1 < k < n/3$, and let $S'$ be a set of size $r < k$. Note that our definition of sum-min() for a singleton $\{u\}$ is $\max_{v \in V} d(u, v)$. Thus picking two points is always better than 1. Thus we may assume that $r \geq 2$, and so we can suppose that $S' = \{u_1, u_2, \ldots, u_r\}$, for $1 < r < k$.

Let us partition $V$ into $V_1, V_2, \ldots, V_r$, where $V_i$ is the set of vertices whose closest neighbor in $S'$ is $u_i$ (i.e., it is the Voronoi partition arising from $S'$). Without loss of generality, assume that $|V_1| \geq |V_2| \geq \cdots \geq |V_r|$. Also, let $d_i$ denote the minimum distance from $u_i$ to the rest of $S'$. We consider two cases:

*Case 1. $|V_1| \geq k$.*

First, consider the set of points $W_1 := \{u_2, u_3, \ldots, u_r\} \cup T$, where $T$ is an arbitrary set of $k - r + 1$ points in $V_1$. We claim that the diversity of $S''$ is at least $(d_2 + d_3 + \cdots + d_r)/2$. This is because for any $i > 1$, every point in $V_1$ is at a distance at least $d_i/2$ from $u_i$ (to see this, consider some $v \in V_1$; we know that $d(v, u_1) \leq d(v, u_i)$, by the definition of $V_1$; thus if $d(v, u_i) < d_i/2$, we must have $d(u_i, u_1) < d_i$, a contradiction).

Second, let $v$ be the $u_i$ that is furthest from $u_1$. Clearly, we have $d(u_1, v) \geq d_1$. Now consider the set of points $W_2 := \{v\} \cup T$, where $T$ is any set of $k - 1$ vertices in $V_1$. From the same argument as above, the diversity of $W_2$ is at least $d_1/2$.

Now one of the sets above must have diversity $\geq (d_1 + d_2 + \cdots + d_r)/4 \geq$ sum-min$(S')/4$. This completes the argument in this case.

*Case 2. $|V_1| < k < n/3$.* In other words, all the sets $V_i$ have size $< k$.

Now let $s$ be the smallest index for which $|V_1 \cup \cdots \cup V_s| \geq k$. Since all the $|V_i|$ are smaller than $k < n/3$, we certainly have $s < r$. Furthermore, we must have $|V_{s+1} \cup \cdots \cup V_r| \geq k$.

Now, define $W_1 := \{u_1, u_2, \ldots, u_s\} \cup T$, where $T$ is an arbitrary set of $k - s$ elements from $V_{s+1} \cup \cdots \cup V_r$, and $W_2 := \{u_{s+1}, \ldots, u_r\} \cup T$, where $T$ is an arbitrary set of $k - s$ elements from $V_1 \cup \cdots \cup V_s$. By the argument above, the diversity of $W_1$ is at least $(d_1 + d_2 + \cdots + d_s)/2$, and that of $W_2$ is at least $(d_{s+1} + \cdots + d_r)/2$.

As before, one of these quantities is at least sum-min$(S')/4$. This completes the proof of the lemma. □

### A.2  Final step – Proof of Lemma 4

We fill in the details of the last step of the lemma – upper bound the second term of (2). Note that

$$\Pr[\chi^{(a)} \wedge \overline{\chi^{(c)}}] \leq \Pr[\chi^{(a)}] \Pr[\overline{\chi^{(c)}} \mid \chi^{(a)}].$$

The second term on the RHS, because the rounding is independent, is at most the probability that the rounding picks at least $k$ pairs *other than* $(i, r)$. Each pair $(j, r')$ is picked with probability $(1 - \epsilon)(1 - e^{-x_{jr'}}) \leq (1 - \epsilon)x_{jr'}$. Thus the expected number of pairs picked (in total) is $\leq (1 - \epsilon)k$. Thus by a standard Chernoff bound, we have

$$\Pr[\overline{\chi^{(c)}} \mid \chi^{(a)}] \leq e^{-\epsilon^2 (1 - \epsilon)k/3} < \epsilon/e,$$

by our choice of $k$. Thus using (2), we have

$$\Pr[\chi_{ir} = 1] \geq \frac{(1 - \epsilon)x_{ir}}{e} - \Pr[\chi^{(a)}] \cdot \frac{\epsilon}{e} \geq (1 - \epsilon)(1 - \epsilon)\frac{x_{ir}}{e}.$$

In the last step, we used the fact that $\Pr[\chi^{(a)}] \leq (1 - \epsilon)x_{ir}$. This completes the proof of the lemma. □

Figure 5: Point $u$ is at a distance $< r_1/2$ from $i_1$ and $< r_2/2$ from $i_2$, and the constraint implies $x_{i_1 r_1} + x_{i_2 r_2} \leq 1$

## A.3 Proof of Theorem 1

First, we illustrate in Figure 5 the additional constraint we add to the linear program.

From Lemma 4, since $\chi_{ir}$ is a 0/1 random variable, it follows that $\mathbb{E}[\chi_{ir}] \geq ((1 - 2\epsilon)/e) \cdot x_{ir}$. Now, using linearity of expectation and Lemma 3, it follows that the expected objective value of the solution returned by the algorithm is at least

$$\mathbb{E}[\sum_{i,r}(r/2)\chi_{ir}] \geq \frac{1 - 2\epsilon}{e} \cdot \sum_{i,r}(r/2)x_{ir} = \frac{1 - 2\epsilon}{2e} \cdot LP_{opt}.$$

I.e., the expected objective value is $\geq (1 - 2\epsilon)/2e$ times the LP-optimum. We now claim that $LP_{opt} \geq \mathsf{opt}$. This will complete the proof, from the above.

To see this, consider the optimal solution to the diversity problem, which is simply a subset $S$ of input points of size $\leq k$. For any $i \in S$, let $d_i$ denote $d(i, S \setminus \{i\})$. Now, set $x_{id_i} = 1$ for all $i \in S$, and the rest of the $x_{ir}$ to 0. For $i$ not in $S$, we always set $x_{ir} = 0$. This solution satisfies all the constraints, as we can easily verify.

The first constraint holds as $|S| \leq k$ and exactly one $x_{ir}$ is set to 1 for each $i \in S$. The second set of constraints hold due to the reasoning given in Section 2. Essentially, suppose the constraint fails for some $u$. Then it means that there exist *at least two* terms on the LHS that are 1. Thus, we have $i, j \in S$ such that $u \in B(i, d_i/2)$, and $u \in B(j, d_j/2)$ (recall the definitions of $d_i, d_j$ above). This means that, by triangle inequality,

$$d(i, j) \leq d(i, u) + d(u, j) < (d_i + d_j)/2 \leq max(d_i, d_j).$$

Thus, if we assume (without loss of generality) that $d_i \leq d_j$, we have that $d(i, j)$ is strictly less than $d_i$, which is a contradiction, because we defined $d_i$ to be $d(i, S \setminus \{i\})$ – the min distance from $i$ to an element of $S \setminus \{i\}$, and $S \setminus \{i\}$ includes $j$.

Now, the cost of this solution ($= \mathsf{opt} = \sum_{i \in S} d_i$) is precisely the LP objective value (since $x_{ir}$ is set to 1 only for $r = d_i$). Thus, since the LP tries to maximize over all feasible solutions, and we have shown that there is a feasible solution of objective value $\mathsf{opt}$, we have that $LP_{opt} \geq \mathsf{opt}$.

# B  General Matroid Constraints

The full LP we use in this algorithm is the following:

$$\text{maximize } \sum_i \sum_{r \in R_i} x_{ir} \cdot r \quad \text{subject to}$$

$$y \in P(\mathcal{M}),$$

$$\sum_{r \in R_i} x_{ir} = y_i \qquad \text{for all } i \in V,$$

$$\sum_{i \in V, r \in R_i : u \in B'(i,r)} x_{ir} \leq 1 \qquad \text{for all } u \in V,$$

$$0 \leq y_i, x_{ir} \leq 1 \qquad \text{for all } i, r.$$

Let us now turn to the analysis of the rounding algorithm. The key component is the following theorem from [7].

**Theorem 4** (Theorem 1.1 in [7]). *Let $(y_1, \ldots, y_n) \in P(\mathcal{M})$ be a fraction solution in a matroid polytope and $(Y_1, \ldots, Y_n) \in \{0,1\}^n$ an integral solution obtained using randomized swap rounding. Then $\mathbb{E}[Y_i] = y_i$, and for any $T \subseteq [n]$, (i) $\mathbb{E}[\prod_{i \in T} Y_i] \leq \prod_{i \in T} y_i$, (ii) $\mathbb{E}[\prod_{i \in T}(1 - Y_i)] \leq \prod_{i \in T}(1 - y_i)$. (Here the expectation is over the randomness of the algorithm.)*

## B.1  Analysis

Let us start by observing that the algorithm returns a feasible solution, i.e., $S \in \mathcal{I}$. This is because, by Theorem 4, $Y \in P(\mathcal{M}) \cap \{0,1\}^V$, and the returned set $S$ is a subset of $\{i : Y_i = 1\}$. Since the feasible sets of a matroid is a down-closed family, we have $S \in \mathcal{I}$.

Let us now turn to the analysis of the objective sum-min$(S)$. As in Section 2, we show that the probability of selecting a point $i \in S$ with a certain radius is proportional to its LP value.

**Lemma 5.** *For any ball $i$ and $r^* \in R_i$, we have*

$$\Pr[i \in S \text{ and } r_i = r^*] \geq x_{ir^*}/4,$$

*where the probability is over the randomness of the generalized-round procedure, i.e., the selection of $Y$ and the radii.*

*Proof.* The statement is clearly true if $x_{ir^*} = 0$. Hence, assume from now on that $x_{ir^*} > 0$. By Theorem 4 the probability that $Y_i = 1$ equals $y_i/2$. Therefore,

$$\Pr[X_{ir^*} = 1] = x_{ir^*}/2,$$

where $X_{ir^*}$ is the indicator variable that $Y_i = 1$ and $r_i = r^*$. Moreover, by condition (ii) of Theorem 4, $\Pr[Y_i = 1 \wedge Y_j = 1] \leq y_i/2 \cdot y_j/2$ for two different centers. In other words, we have $\Pr[Y_j = 1 \mid Y_i = 1] \leq y_j/2$ and therefore $\Pr[X_{jr} = 1 \mid X_{ir^*} = 1] \leq x_{jr}/2$.

Also note that $i$ is in the returned set $S$ with $r_i = r^*$ if $X_{ir^*} = 1$ and $X_{jr} = 0$ for all $r \geq r^*$ such that $i \in B'(j,r)$. Thus, by the union bound, $\Pr[i \in S \text{ and } r_i = r^*]$ is at least

$$\Pr[X_{ir^*} = 1] \left(1 - \sum_{j,r \in R_j : r \geq r^* \text{ and } i \in B'(j,r)} \Pr[X_{jr} = 1 \mid X_{ir^*} = 1]\right)$$

$$\geq x_{ir^*}/2 \left(1 - \sum_{j,r \in R_j : r \geq r^* \text{ and } i \in B'(j,r)} x_{jr}/2\right)$$

$$\geq x_{ir^*}/4,$$

where the last inequality follows from the LP constraint $\sum_{j,r \in R_j : i \in B'(j,r)} x_{jr} \leq 1$. $\qquad \square$

The bound sum-min$(S) \geq$ opt$/8$ now follows from linearity of expectation in the same way as in Section 2 (Lemma 3). This completes the proof of Theorem 2.

## C Coverage Objective in Addition to Diversity

We start by discussing the statement of Theorem 3. Note that the guarantees of the theorem are all in expectation. Thus we may, with some probability, output a set of size $> k$. It could also happen that the solution we output has only a high value for diversity or has good coverage but not both (because guarantees are in expectation). We do not expect to see such pathological cases in practice, but it is one drawback of randomized rounding. We leave it as an interesting open problem to obtain a bi-criteria approximation.

Let us now describe the algorithm. The overall idea is to write a linear program which *combines* the linear program for diversity maximization with the natural one for maximizing coverage, and then round via randomized rounding. Formally, in addition to the $x_{ir}$ variables from Section 2, the LP also has variables $z_j$ for $j \in [m]$, which are supposed to indicate if element $j$ is covered.

$$\text{maximize} \sum_i \sum_{r \in R_i} x_{ir} \cdot r \quad \text{subject to}$$

$$\sum_{i \in V, r \in R_i} x_{ir} \leq k,$$

$$\sum_{i \in V, r \in R_i : u \in B'(i,r)} x_{ir} \leq 1 \qquad \text{for all } u \in V,$$

$$\sum_{j \in [m]} z_j \geq C,$$

$$z_j \leq \sum_{(i,r) : B'(i,r) \ni j} x_{ir} \qquad \text{for all } j \in [m],$$

$$0 \leq z_j, x_{ir} \leq 1.$$

Note that the last three constraints are the ones that correspond to the coverage problem, while the first two are the ones we had for diversity in Section 2. Another technical point is that we assumed knowledge of the parameter $C$ when formulating the LP. We can assume this because we can try various guesses for $C$ (in increasing multiples of $(1 + \delta)$, for a small constant $\delta$), and work with the largest $C$ for which the LP is feasible.

**Feasibility of the LP.** Consider any solution $S^*$ with size $\leq k$, and suppose it has a coverage objective at least $C$. Then we can construct a feasible solution to the LP as in Section 2: we set $x_{ir} = 1$ precisely when $i \in S^*$ and $r = \min d(i, S^* \setminus \{i\})$. The value of $z_j$ is set to 1 for all $j \in \cup_{i \in S^*} C_i$.

The algorithm is a simplified version of the one from Section 2. Here the rounding is done with probability simply $x_{ir}/2$, and there is no ensuring that the solution has size $\leq k$.

---

procedure round-coverage$(x, z)$     // LP solution $(x, z)$.
 1: Initialize $\mathcal{S} = \emptyset$.
 2: Add $(i, r)$ to $\mathcal{S}$ with probability $x_{ir}/2$ (independent of the other point-radius pairs).
 3: If $(i, r) \neq (j, r') \in \mathcal{S}$ such that $r \leq r'$ and $i \in B'(j, r')$, remove $(i, r)$ from $\mathcal{S}$.
 4: Return the first coordinates of the pairs in $\mathcal{S}$.

---

The technical core in the analysis is to show that the *removal* phase (step 3) does not cause a significant loss to the coverage objective, in expectation.

We now show how to analyze procedure round-coverage and prove Theorem 3.

From the rounding, it is clear that part (1) of Theorem 3 holds. Now let us define $\chi_{ir}$ to be 1 iff (a) $(i, r)$ is selected in step 2, and (b) $(i, r)$ is not removed in step 3. Analyzing $\Pr[\chi_{ir} = 1]$ is now much simpler.

**Lemma 6.** *In the notation above,* $\Pr[\chi_{ir} = 1] \geq x_{ir}/4$.

This lemma, together with Lemma 3 implies that part (2) of the statement of Theorem 3 holds.

*Proof.* As before, let $T$ be the set of all (point, radius) pairs $(j, r')$ such that $(i, r) \neq (j, r')$, $i \in B'(j, r')$, and $r' \geq r$. Now, the condition (b) in the definition of $\chi_{ir}$ is precisely the same as saying that none of the pairs in $T$ are picked in step 2. Thus

$$\Pr[\chi_{ir} = 1] = \frac{x_{ir}}{2} \prod_{(j,r') \in T} \left(1 - \frac{x_{jr'}}{2}\right).$$

Since $\sum_{(j,r') \in T} x_{jr'} \leq 1$, we have that the product term above is $\geq 1/2$, implying the lemma. $\square$

We thus need to analyze the coverage objective. This turns out to be a bit tricky because of the *removal* phase (step 3).

**Lemma 7.** *Fix some $j \in [m]$ and consider the algorithm above. The probability that it outputs a solution that* covers $j$ *is at least* $z_j/16$.

*Proof.* Let us denote by $N_j := \{(i, r) : C_i \ni j\}$. We will now analyze what happens the elements of $N_j$ as we execute the algorithm. Let $X_{ir}$ be the indicator of event that the algorithm adds $(i, r)$ to $S$ in step 2, and $Y_{ir}$ the indicator that it does *not* discard it in step 3. Thus $X_{ir}, Y_{ir}$ are $0/1$ random variables, and we have $\Pr[X_{ir} = 1] = x_{ir}/2$ by definition, and $\Pr[Y_{ir} = 1] \geq x_{ir}/4$ from our analysis earlier. We need to lower bound the probability of the event $\sum_{(i,r) \in N_j} Y_{ir} > 0$. The proof proceeds by analyzing two cases:

*Case 1.* We have $\sum_{(i,r) \in N_j} x_{ir} \leq 2$.

In this case we show that, in fact, the probability that the algorithm picks *precisely one* $(i, r)$ in $N_j$ in step 2 is at least $z_j/16$.

This is because of the following: consider some $(i, r) \in N_j$, and let $U_{ir}$ denote the event that (a) $X_{ir} = 1$, (b) $X_{i'r'} = 0$ for all other $(i', r') \in N_j$, and (c) $X_{i'r'} = 0$ for all other balls $i'$ that could cause $Y_{ir} = 0$ (in step 3 of the algorithm). Now, if event $U_{ir}$ occurs, we must have $Y_{ir} = 1$ (because $(i, r)$ is picked, and it does not get removed). Furthermore, the events $U_{ir}$ are mutually disjoint for different $(i, r) \in N_j$ (because we insist in (b) that $X_{ir} = 1$ and the rest are 0). Thus the *sum* of this probability over $(i, r) \in N_j$ will be a lower bound for $\Pr[\sum_{(i,r) \in N_j} Y_{ir} > 0]$.

Let us thus compute the probability of $U_{ir}$ for a fixed $(i, r) \in N_j$. We call the set of possible $(i', r')$ in (b) and (c) above, the *blocking* set for $(i, r)$, and denote it by $\mathsf{Block}(i, r)$. Hence we have

$$\Pr[\, X_{ir} = 1 \ \wedge \ X_{i'r'} = 0 \text{ for all } (i', r') \in \mathsf{Block}(i, r)]$$
$$= \frac{x_{ir}}{2} \prod_{(i',r') \in \mathsf{Block}(i,r)} \left(1 - \frac{x_{i'r'}}{2}\right)$$

From the reasoning in Lemma 6, we have that the sum of $x_{i'r'}$ values of the pairs arising from condition (c) above is at most 1, and by our assumption for Case 1, we have $\sum_{(i',r') \in N_j} x_{i'r'} \leq 2$. Thus, we have $\sum_{(i',r') \in \mathsf{Block}(i,r)} x_{i'r'} \leq 3$. Subject to this (and the fact that $0 \leq x_{i'r'} \leq 1$), the minimum value of $\prod_{(i',r') \in \mathsf{Block}(i,r)} \left(1 - \frac{x_{i'r'}}{2}\right)$ occurs when the $x_{i'r'}$ are as spread out as possible, i.e., when precisely three of them are 1 and the rest are 0. Thus the product is $\geq 1/8$, implying that

$$\Pr[\, X_{ir} = 1 \ \wedge \ X_{i'r'} = 0 \text{ for all } (i', r') \in \mathsf{Block}(i, r)] \geq x_{ir}/16.$$

Combining this with the discussion above, we have that $\Pr[\sum_{(i,r) \in N_j} Y_{ir} > 0] \geq \sum_{(i,r) \in N_j} x_{ir}/16$, which in turn is at least $z_j/16$. This completes the proof in this case.

*Case 2.* We have $\sum_{(i,r) \in N_j} x_{ir} > 2$. In this case, we may assume that $z_j = 1$, and we will show that the probability that at least one element of $N_j$ is selected is at least $1/16$.

Let us write $C = \sum_{(i,r) \in N_j} x_{ir}$. Recalling the definitions of $X_{ir}$ and $Y_{ir}$, we have, from linearity of expectation and Lemma 6,

$$\mathbb{E}[\sum_{(i,r) \in N_j} Y_{ir}] \geq C/4.$$

Let $q$ denote the probability that $\sum_{(i,r)\in N_j} Y_{ir} > 0$. We want to show that $q \geq 1/16$. By definition, the above can be written as

$$q \cdot \mathbb{E}[\sum_{(i,r)\in N_j} Y_{ir} \mid \sum_{(i,r)\in N_j} Y_{ir} > 0] \geq C/4.$$

This implies (since $X_{ir}$ is always $\geq Y_{ir}$), that

$$\mathbb{E}[\sum_{(i,r)\in N_j} X_{ir} \mid \sum_{(i,r)\in N_j} Y_{ir} > 0] \geq C/4q.$$

The rest of the argument proceeds by showing that if $q < 1/16$, this cannot happen. In particular, since $\sum_{(i,r)\in N_j} X_{ir}$ is a sum of independent Bernoulli random variables with mean equal to $C/2$, it is quite concentrated around $C/2$, and if $q$ is small enough, we cannot have the desired expectation to be $\geq C/4q$.

Let us now make the intuition formal. For convenience, let us denote by $Q$ the event that $\sum_{(i,r)\in N_j} Y_{ir} > 0$. We start by noting that for any $0 < t \leq 1$, since $X_{ir}$ is 1 with probability $x_{ir}/2$ and 0 otherwise, that

$$\mathbb{E}[e^{t(X_{ir}-x_{ir}/2)}] \leq e^{t^2\mathbb{E}[(X_{ir}-x_{ir}/2)^2]} \leq e^{t^2 x_{ir}/2}.$$

(This is the standard argument used to prove Chernoff style tail bounds.) Since the $X_{ir}$ are independent, this implies that

$$\mathbb{E}[e^{t(\sum_{(i,r)\in N_j} X_{ir})-tC/2}] \leq e^{t^2 C/2}.$$

Now, since the exponential function is non-negative and convex, we have that

$$q \cdot e^{t\cdot\mathbb{E}[\sum X_{ir}|Q]-tC/2} \leq \mathbb{E}[e^{t(\sum X_{ir})-tC/2}] \leq e^{t^2 C/2},$$

where the summations are over $(i,r) \in N_j$, as before. Let us write $E := \mathbb{E}[\sum_{(i,r)\in N_j} X_{ir}|Q]$. Taking logs, we have

$$-\log(1/q) + tE - tC/2 \leq t^2 C/2 \implies E \leq C/2 + \frac{\log(1/q)}{t} + tC/2.$$

Setting $t = 1$, we get $E \leq C+\log(1/q) = C\left(1 + \frac{\log(1/q)}{C}\right)$. This is a contradiction if $1+\frac{\log(1/q)}{C} < 1/4q$, which can easily be seen to hold if $C > 2$ and $q \leq 1/16$.

This shows that in both the cases, the probability of covering element $j$ is at least $z_j/16$, completing the proof. □

## D Hardness beyond factor $1/2$

We show that assuming the hardness of the *planted clique problem* (which is widely believed), it is impossible to obtain an approximation factor better than $1/2$ for diversity maximization under both the *sum-sum* and the *sum-min* objectives. We note that our hardness results hold whether we insist on returning a set of size precisely $k$, or if we allow sets of size $\leq k$.

**Finding planted cliques.** The well known *planted clique* problem, introduced by Karp [13], asks for an algorithm that can distinguish, with probability $\geq 3/4$, between graphs drawn from the following two distributions ($p$ and $\delta$ are parameters):

$\mathcal{D}_1 : G$ is drawn from $\mathcal{G}(n,p)$.

$\mathcal{D}_2 : G$ is drawn from $\mathcal{G}(n,p)$, and then a clique of size $n^{1/2-\delta}$ is *planted* (i.e., a set of vertices of this size is chosen at random and all edges between those vertices are added).

For any constants $p \in (0,1)$ and $\delta > 0$, we do not know of polynomial time algorithms that can solve the distinguishing problem above, and the *planted clique conjecture* states that it is impossible to do so. Recently, there has been evidence for the planted clique conjecture, in the form of lower bounds for linear and semidefinite programming relaxations, as well as lower bounds for broad classes of algorithms (see [9]). Let us now state our results formally.

We start by noting that the planted clique conjecture is usually stated with $p = 1/2$, but it is believed for all constants $p$ and $\delta$, so our statement is slightly more general. Our result is formally stated as follows.

**Theorem 5.** *Assuming the planted clique conjecture, for any constant $\epsilon > 0$, it is impossible to approximate (a) the sum-sum diversity, and (b) the sum-min diversity to a factor better than $(2 - \epsilon)$ in polynomial time.*

We now prove Theorem 5, by a simple reduction. We first note that the planted clique conjecture immediately implies (by taking the complement of the graph) that for any $0 < p < 1$, it is impossible, in polynomial time, to distinguish between a graph $G \sim \mathcal{G}(n, p)$ (call this $\mathcal{D}_1'$), and a graph with a planted *independent set* of size $n^{1/4}$ (call this $\mathcal{D}_2'$). Let us also fix $\delta = 1/4$). Now given a graph $G$, denote the shortest path metric by $d$. We now have the following lemmas:

**Lemma 8.** *Let $G \sim \mathcal{D}_2'$ (i.e. the planted case), and set $k = n^{1/4}$. Then we have*

$$\mathsf{opt}_1(k) \geq 2k, \qquad and \qquad \mathsf{opt}_2(k) \geq 2k(k-1).$$

*Proof.* The intuition is that the vertices of the planted independent set are all *far* from each other. Formally, it is easy to check that the contribution of every vertex is $\geq 2$ for the sum-min objective, and $\geq 2(k-1)$ for the sum-sum objective. This implies the lemma. $\square$

**Lemma 9.** *Let $G \sim \mathcal{D}_1'$ (i.e., $G(n, p)$ without planting), and set $k = n^{1/4}$. Then for any $k' \leq k$, we have*

$$\mathsf{opt}_1(k') \leq (1 + o(1))k, \qquad and \qquad \mathsf{opt}_2(k') \leq (2 - p + o(1))k(k-1)$$

*with probability $1 - 1/n^2$, where the $o(1)$ term hides factors that tend to $0$ with $n$.*

In the case of $\mathsf{opt}_1$, for any chosen vertex, it is extremely unlikely that *none* of the other vertices is connected to it (at least for a typical chosen vertex), thus it contributes $1$ to the objective. In the case of $\mathsf{opt}_2$, we expect it to be at a distance $1$ to a good fraction of the chosen vertices, which gives the desired bound. We formalize this argument shortly, but first, we note that this implies the theorem.

*Proof of Theorem 5.* Using Lemmas 8 and 9, and by setting $p$ to be $1 - \epsilon$, our main theorem follows. $\square$

### D.1 Proof of Lemma 9

We start by noting that with probability $1 - \exp(-\Omega(n))$, we have that $d(u, v) \leq 2$ for any $u, v \in V(G)$. This is simply because the expected number of common neighbors is $p^2 n$, and since $p = \Omega(1)$, the probability that this is $0$ is $\exp(-\Omega(n))$. We can take a union bound over all pairs, and from now, assume that $d(u, v) \leq 2$ for all pairs $u, v$.

Thus for any set $S \subseteq V$ of size $< k/2$, we have $\mathsf{sum\text{-}min}(S) \leq 2|S| < k$ and $\mathsf{sum\text{-}sum}(S) \leq 2|S|(|S| - 1) < k(k - 1)$. This means that we may assume that sets maximizing sum-min and sum-sum are of size $> k/2 = \omega(\log n)$.

The next observation is that for any two sets of nodes $R, T$ of sizes $r \geq t$ respectively, the number of edges is at least $rtp^2$, and thus the probability that there are no edges is $\exp(-\Omega(rt))$. If $T = c \log n$ for an appropriately large $c$, this probability is smaller than $n^{-4(r+t)}$, implying that with $1/n^{\omega(1)}$ probability, there are no such sets $R, T$.

Now consider any set $S \subseteq V(G)$ of size $\geq k/2$. From the above argument, the size of $T = \{s \in S : \Gamma(s) \cap S = \emptyset\}$ is at most $c \log n = o(1)|S|$ (since $|S| \geq k/2$). This immediately implies that $\mathsf{sum\text{-}min}(S) \leq (1 + o(1))|S|$.

A final observation for sets $S$ of size $\geq k/2$ is that the expected number of edges is $|S|(|S| - 1)p/2$, and thus the probability that the number is $< |S|(|S| - 1)p(1 - \epsilon)/2$ is at most $\exp(-\Omega(\epsilon^2|S|^2 p))$. Setting $\epsilon = 1/\log n$, and recalling our choice of $k$, we see that this probability is $< 1/n^{\omega(k)}$, and thus we can take a union bound over all $S$ of size between $k/2$ and $k$. Thus with probability $1 - 1/n^{\omega(1)}$, all sets $S$ of size between $k/2$ and $k$ have at least $|S|(|S| - 1)p(1 - o(1))/2$ edges. This (along with the observation that any two vertices are at distance at most 2) immediately implies the bound on $\mathsf{sum\text{-}sum}(S)$. This completes the proof of the lemma. $\square$

# E   Bad Instances for Greedy and Local Search

We give an instance in which the greedy and local search algorithms have an approximation ratio $O(1/\sqrt{k})$ for the sum-min objective.

**Description of the Instance.**   We have $\sqrt{k}$ sets of vertices $S_1, S_2, \ldots, S_{\sqrt{k}}$, each of size $k$. For any two vertices $u, v$ in different sets (i.e., $u \in S_i$ and $v \in S_j$, with $i \neq j$), we have $d(u, v) = 4$. Each $S_i$ has $\sqrt{k}$ special vertices $T_i$, that are at a distance 1 from one another (i.e., for all $u, v \in T_i$, $d(u, v) = 1$). Finally, around each vertex $u \in T_i$, for all $i$, there are $\sqrt{k}$ vertices $W_u$ that are at a distance 0 to $u$, and 0 to one another. (If we wish to avoid zero distances, we can obtain the same effect with a distance $\epsilon$, for a small enough $\epsilon$.) For vertices $i \in W_u$ and $j \in W_v$, we have $d(i, j) = d(u, v)$.

This is an instance with $O(k\sqrt{k})$ vertices.

**Optimal solution.**   We wish to pick $k$ vertices so as to maximize sum-min diversity. Choosing $T_1 \cup T_2 \cup \cdots \cup T_{\sqrt{k}}$ will give a total diversity of $k$ (every vertex contributes 1 to the sum).

Now, what vertices does the greedy algorithm pick?

**Greedy algorithm.**   In the first $\sqrt{k}$ steps, the algorithm will pick one vertex from each of the $S_i$. Let $u_i$ be the vertex it picked from $S_i$. Next, the algorithm will pick a second vertex from one of the $S_i$, say $S_1$. Now, let us consider the choices in the next step. If the algorithm picks a second vertex from an $S_i$, with $i \neq 1$, then the contribution of $u_i$ to the summation changes from 4 to $\leq 1$, and the additional contribution of the newly added vertex is at most 1, thus there is a net drop $\geq 2$. On the other hand, if a vertex is chosen from $S_1$, the net drop is $\leq 1$, so it is preferable to pick a vertex from $S_1$. This reasoning holds for all the remaining steps of the algorithm. Thus the greedy algorithm will end up picking $k - \sqrt{k} + 1$ vertices from $S_1$, and one vertex each from the other $S_i$. This gives a total sum-min objective of $\leq 4\sqrt{k}$.

**Local search.**   Consider the solution chosen by the greedy algorithm. It is locally optimum, because swapping in any vertex from $S_i$ ($i \neq 1$) will only result in a drop in the sum-min objective value.

This shows that both the greedy and the local search algorithms have an approximation factor of $O(1/\sqrt{k})$ in the worst case, thus proving Lemma 1.