[Reviews · NeurIPS 2016]

Reviewer 1

Summary

The paper considers the problem of selecting M diverse elements of a finite set, given that those are elements of a metric space and therefore a distance between elements is defined. More precisely, the work focuses on the so called 'sum-min diversity', which has attractive practical properties. The problem of finding the M-tuple of diverse points with respect to such measure is known to be NP-hard and the best known algorithm provides an approximation factor of O(1/log n), where n is the size of the set. Authors propose a linear programing approximation of the problem and a probabilistic rounding procedure to obtain a binary vector defining the diverse M-tuple. The expected output of this probabilistic algorithm provides a constant factor approximate solution for the problem.

Qualitative Assessment

Although the provided novel algorithm looks impressive both from the theoretical prospective and in the experimental comparison, its substantiation has quite some room for improvement. The major point is the proof of Theorem 1: - it is unclear how the proof of the theorem follows from Lemmas 3 and 4, since none of these lemmas is related to the optimal solution of the considered diversity problem. I assume that the missing proposition is the one, which would establish connection between the considered linear program in lines 153-154 (by the way, it is very uncomfortable that the main formulation is not numbered and therefore can not be easily referenced) and the diversity problem. I believe that this connection may have the following format: if the linear program is equipped with integrality constraints (which is, all variables x_{ir}\in {0,1}), the resulting ILP is equivalent to the considered diversity problem. Indeed, the proof of such a proposition is not obvious for me as well. Without clarification of this point I would assume a major flaw in the proof and, therefore, absence of a theoretical contribution. == post-rebuttal update== The authors addressed my main issue - the proof of Theorem 1. Although I agree to R7 about experimental evaluation, I believe the paper can be accepted provided that the mentioned experimental comparison is included into the paper.

Confidence in this Review

2-Confident (read it all; understood it all reasonably well)


Reviewer 2

Summary

The paper studies LP methods for obtaining approximation algorithms for picking diverse sets of points in metric spaces. The authors concentrate on the sum-min diversity function.

Qualitative Assessment

The paper is written very clearly and very cogently. It is a pleasure to read. The contribution of the paper is mostly theoretical but the authors provide some nice experimental evaluation on real datasets. Interestingly, experimental results suggest that the approximation constant is better 1/8, the theoretical approximation constant.

Confidence in this Review

2-Confident (read it all; understood it all reasonably well)


Reviewer 3

Summary

The paper considers the problem of finding a diverse subset in a metric space. The paper proposes Linear Relaxation based method for sum-min diversity objective. The authors provide proof of constant factor approximation for the new approach. Experimental evaluation of the new method is provided.

Qualitative Assessment

Technical quality. I understand that greedy and local search algorithms for sum-min objective have a bad approximation ratio. However, in order to prove that the experiments in the paper are sufficient, sum-min greedy and sum-min local search algorithms have to be used for experimental evaluation as the baselines. Novelty and potential impact. The paper sounds novel to me and it can have a great impact. Clarity. The paper is well written. I would suggest to include some graphical explanation for eq. (1) to further improve it. * post-rebuttal update * I checked the proof of Theorem 1 from the authors' rebuttal. In my opinion, it is correct. It should be included in the paper. Overall, I really like the theoretical contribution of the paper. My question about sum-min greedy and sum-min local search baselines in the experimental evaluation is not addressed properly in the rebuttal. I believe that the provided experiments are sufficient for the new method evaluation only if it outperforms these baselines, otherwise, these experiments only show the known fact that the sum-min objective is better than the sum-sum one for the diversity problem.

Confidence in this Review

2-Confident (read it all; understood it all reasonably well)


Reviewer 4

Summary

This paper presents a linear programming framework for diverse subset selection under matroid constraints. The authors propose simple randomized algorithms to select sum-min diverse elements according to the solution of a linear program. The paper provides approximation guarantees as well as hardness results which state the algorithms are nearly optimal in many cases. Experimental results show that both LP relaxation and the LP framework select diverse subsets on real datasets.

Qualitative Assessment

This paper flows very well. Ideas are presented logically, clearly, and concisely. My only comments on presentation are that the paper ends abruptly, and if possible I suggest moving the technical statement on planted clique hardness to the main text. This work appears to be a novel mix of linear programming and matroid constrained subset selection with substantial theoretical contribution. However, the datasets used for experiments are somewhat small. Large scale examples (synthetic if necessary) would be useful, as well as a comparison to existing feature selection methods. I am also curious about the algorithms' performance under metrics commonly used in related work, including running time.

Confidence in this Review

1-Less confident (might not have understood significant parts)


Reviewer 5

Summary

Authors propose an augmented linear programing algorithm for finding collection of points with optimal diversity. Authors first discuss different objective functions for measuring diversity and justify the use of the sum-min measure. Then, they propose a linear programming formulation for which they prove a hardness result. This algorithm is applied to benchmarks - but the paper's outcome is not discussed.

Qualitative Assessment

The paper provides with a clear solution to a well defined problem. Though I am not an expert, I would suggest some modifications to make the contributions clearer. In particular, the algorithm appears to have important applications and seems better than other methods as shown in your results. First, the methods for the presentation of the results are to succint and do not allow to really understand how you perform the experiments. In particular chosing a (non-whitened) euclidian distance between 32x32 image patches to classify objects appears to be a rather bad choice for image descriptors. Moreover, the assesment of results is not clear to me, for instance in terms of number of covered clusters (relative to the number of objects?). Perhaps a first result on synthetic data on the different methods whould allow to illustrate your main points (such that corners do not get selected) and to sort out the respective advantages of using the sum-min and of your method. I would also suggest that you conclude your paper by a rapid discussion.

Confidence in this Review

1-Less confident (might not have understood significant parts)